# ATM: Improving Model Merging by Alternating Tuning and Merging

## Abstract

Model merging has recently emerged as a cost-efficient paradigm for Multi-task Learning (MTL). Among merging solutions, Task Arithmetic (Ilharco et al., 2022) stands out for its simplicity and effectiveness. In this paper, we start by motivating the effectiveness of task vectors with their relation to multi-task gradients. We show that in the single epoch scenario, task vectors are exactly equivalent to gradients obtained by performing gradient descent in a multi-task setting, and still approximate the latter with further epochs. We further strengthen the explanation by showing that task vectors work best when equality is maintained and motivate their effectiveness in the general case by showing that most of the contribution in the total update is determined by the gradient of the first epoch. Guided by this parallel, we propose viewing model merging as a single step in an iterative process that **A**lternates between **T**uning and **M**erging (ATM). Acting as a midpoint between model merging and multi-task gradient descent, ATM obtains state-of-the-art results with the same data and computing requirements. We first extensively evaluate our approach under diverse settings, demonstrating state-of-the-art performance, leading by an accuracy of up to $19\%$ in computer vision and $20\%$ in NLP over the best baselines. We then motivate its effectiveness empirically, showing increased orthogonality between task vectors and, theoretically, proving it to minimize an upper bound to the loss obtained by finetuning jointly on all tasks.

## 1 Introduction

The pretrain-and-finetune paradigm has become the standard approach for numerous deep learning tasks. In this framework, a model pretrained on large-scale unlabeled data is adapted to a specific downstream task with minimal tuning. However, when addressing multiple tasks, a key limitation is the need to store separate finetuned models for each task. Model merging addresses this issue by combining task-specific models into a single model capable of handling all tasks. This approach significantly reduces storage costs, as the unified model's size remains comparable to that of a single task model, regardless of task count. Among numerous model merging methods, *task arithmetic* (Ilharco et al., 2022) stands out for its simplicity and effectiveness. Given a pretrained model $\theta_0$ and a model $\theta_i$ finetuned on task $t_i$, the task vector $\tau_i$ is the subtraction of the pretrained weights from the finetuned ones (i.e. $\tau_i = \theta_i - \theta_{\text{init}}$). For multi-task learning with $n$ tasks, task arithmetic computes the sum of the $n$ task vectors, properly scales it with a coefficient $\alpha$, and adds the resulting vector to the pretrained model.

In this paper, we motivate the effectiveness of task arithmetic by relating task vectors with gradients of the average loss over all the tasks. In particular, we show that when a model is finetuned for a single epoch using Gradient Descent, the corresponding task vector is exactly the additive inverse of the gradient of the loss, rescaled by the learning rate. Analogously, the multi-task vector obtained by summing the task vectors is equivalent to the additive inverse of the gradient of the average loss across all tasks. With this perspective, task addition is equivalent to a step of gradient descent on the sum of the average losses of the tasks. When the finetuning is performed for several epochs, the equality is replaced by an approximation with an error dependent on the learning rate. We further show that, while the single-epoch-finetuning assumption is violated in practice, the analogy to gradients can still explain why task vectors work in the first place. In fact, we show that, if we consider the trajectory in the parameter space followed by the model during fine-tuning, the greatest contribution in terms of gradient norm is given by the first epoch. When this does not hold, we often

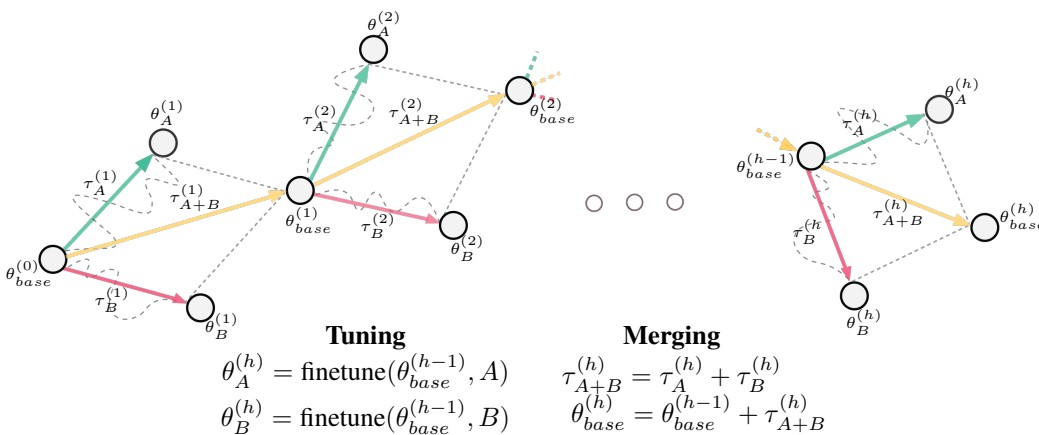

**Tuning**

$$\theta_A^{(h)} = \text{finetune}(\theta_{base}^{(h-1)}, A)$$

$$\theta_B^{(h)} = \text{finetune}(\theta_{base}^{(h-1)}, B)$$

**Merging**

$$\tau_{A+B}^{(h)} = \tau_A^{(h)} + \tau_B^{(h)}$$

$$\theta_{base}^{(h)} = \theta_{base}^{(h-1)} + \tau_{A+B}^{(h)}$$

Figure 1: An illustration of the ATM framework up to the iteration $h$ with $|T| = 2$ tasks (A and B). Starting with the pretrained model $\theta_{base}^{(0)}$ as the base model, the **FT** step consists of finetuning it separately on the $|T|$ tasks and the **Merge** step performs task vector aggregation and applies the multi-task task vector to the current base model, resulting in the next-iteration multi-task model. The process iterates with the resulting model at each iteration as the new base model for the next, until predefined iteration $h$ or when some predefined condition is met. We observe increased task vector orthogonality as the ATM iteration grows.

find the gradients from the subsequent epochs to be aligned with the previous one, confirming that the direction is indeed dictated by the latter.

In this view, aggregation and merging in task arithmetic correspond to a noisy step of gradient descent when finetuning on the union of tasks, using as loss the sum of the average losses of the tasks. In practice, this means that the one-step nature of these techniques likely results in overshooting the multi-task minimum, as they would be actually tackling gradient descent over a multi-task dataset with a single noisy step, where the multiplicative factor that is optimized over the validation set is, in fact, the learning rate. Building on these insights, we overcome the limitations of traditional one-step task vector applications by introducing **A**lternating **T**uning and **M**erging (ATM) - a novel multi-task model merging framework that generalizes task arithmetic by iteratively alternating between finetuning and merging, enabling a more gradual and refined integration of task-specific knowledge. Given a compute budget of $b$ epochs per task, traditional methods finetune each task for $b$ epochs in a single pass. In contrast, ATM distributes the budget across $k$ iterations, with each iteration performing $\frac{b}{k}$ epochs of finetuning followed by task vector aggregation. The unified model from each iteration serves as the starting point for the next. After $k$ iterations, the final unified model is deployed and evaluated. Notably, ATM is agnostic to the merging framework, allowing the integration of any interference-resolution techniques during the merge step to enhance performance. In general, ATM significantly reduces time overhead compared to current baselines, and extensive experiments in computer vision and NLP show that it achieves state-of-the-art results without requiring hyperparameter tuning.

Our analysis of ATM unveils intriguing properties that shed light on its effectiveness. First, ATM task vectors exhibit a higher degree of orthogonality compared to baseline methods. We further prove that, in the simplified case where gradient descent is used to update the model, ATM minimizes the loss obtained by training jointly on all tasks. Our code, together with experiment configurations and checkpoints, is available for reproducibility purposes [1]

Our contribution is four-fold and is summarized as follows:

- We expose that task vectors, under certain conditions, are either equivalent to or approximate the gradients of loss of the corresponding tasks.

---

[1]Link concealed to preserve anonymity.

- We point out that existing merging frameworks adopt one-shot merging, which likely over-shoots the multi-task optimum, especially when task vectors exhibit large norms.

- We propose Alternating Tuning and Merging (ATM), a novel state-of-the-art model merging framework that generalizes task vector arithmetic. Thanks to its flexibility, ATM can readily integrate any interference-resolution framework, requiring no additional overhead.

- We empirically and mathematically motivate the effectiveness of ATM by showing increased orthogonality between task vectors obtained through ATM as compared to standard ones. We additionally prove that ATM reduces the loss of a multi-task model finetuned on the union of the datasets.

## 2 RELATED WORK

**Mode connectivity and model merging** Mode connectivity studies the weights that characterize local minima on the loss landscape. Frankle et al. (2020) investigated linear mode connectivity in models briefly trained from identical starting points, while Entezari et al. (2022) speculated that all models converge to a shared basin once neuron permutations are resolved. Leveraging these insights, permutation-based model merging aims to combine diverse models into a unified one, aiming to inherit their capabilities without the overhead associated with ensembling. Singh & Jaggi (2020) proposed an optimal-transport weight-matching method, while Git Re-Basin (Ainsworth et al., 2022) proposed three novel matching methods acting both on weights and activations, with REPAIR (Jordan et al., 2023) demonstrating significant barrier reduction through activation renormalization. Most recently, Navon et al. (2023) proposed merging models in the embedding space of deep neural networks, while Crisostomi et al. (2024) proposed a cycle-consistent matching procedure for improved merging. When models share the same pretrained initialization, Wortsman et al. (2022) propose fusing them via a simple average. Jolicoeur-Martineau et al. (2023) propose to merge models by pushing them towards the population mean to ensure stability. RegMean Jin et al. (2022) and Fisher-weighted averaging Matena & Raffel (2021) fall under the regime of weighted averaging, where the weights are optimized according to some criteria. Daheim et al. shed light on the positive relation between post-averaging multi-task performance and the gradient mismatch between the constituent models. Finally, Choshen et al. (2022) even proposed model merging in replacement of pertaining. They argue that pretrained checkpoints are not always the optimal starting point for further finetuning, and a model resulting from merging finetuned models can be a better starting point than any of its constituents.

**Task vectors** Task vector-based merging (Ilharco et al., 2022) finetunes a pretrained model on different tasks to obtain task vectors (differences between finetuned and original checkpoints). Arithmetic operations on these vectors enable forgetting, analogy learning, and multi-task learning. Several works aim to improve task vector merging by reducing task interference (Deep et al., 2024; Wang et al., 2024; Huang et al., 2024). Some methods include sparsifying task vectors or finetuning only lottery tickets (Panda et al., 2024). TIES-merging (Yadav et al., 2023) merges vectors by pruning, selecting a unified sign vector, and merging disjointly, while Model Breadcrumbs (Davari & Belilovsky, 2023) prunes both small and large-magnitude weights. DARE Merging (Yu et al., 2023) randomly masks out a portion of weights and scales up the rest. AdaMerging (Yang et al., 2023) optimizes aggregation coefficients, while Yang et al. (2024) propose task-specific modules for test-time adaptation. Ortiz-Jimenez et al. (2024) introduce the concept of weight disentanglement and propose finetuning in the tangent space. Unlike these one-shot methods, we introduce the perspective of iterative model merging, evolving a base model towards a better multi-task performance.

## 3 TASK VECTORS AS GRADIENTS

In this section, we show that task vectors are tightly related to the gradients of the loss over the union of the tasks.

**Theorem 3.1.** *Let* $\left\{\theta_t^{(k)}\right\}_{t=1}^{|T|}$ *be a set of models obtained by finetuning* $\theta_{base}$ *for* $k$ *epochs over the set of tasks* $T$ *via Gradient Descent (GD) with a learning rate of* $\eta$, *where fine-tuning over task* $t \in T$ *minimizes* $\overline{L}_t(\theta) = \frac{1}{n_t} \sum_{i=1}^{n_t} \ell(x_i, y_i, \theta)$. *Let moreover* $\left\{\tau_t^{(k)}\right\}_{t=1}^{|T|}$ *be a set of task vectors,*

with each $\tau_t^{(k)} = \theta_t^{(k)} - \theta_{base}$. Let $\tau_{MT}^{(k)}$ be the multi-task vector $\tau_{MT}^{(k)} = \sum_{t \in T} \tau_t^{(k)}$. Finally let $\theta_{MT}^{(k)}$ be the model obtained minimizing the loss $\sum_{i=1}^{|T|} \overline{L}_i$ for $k$ epochs with GD using learning rate equal to $\alpha\eta$. It holds that

$$\tau_{MT}^{(1)} = -\eta \nabla \sum_{t \in T} \overline{L}_t(\theta_{base}) \tag{1}$$

$$\tau_{MT}^{(k)} = -\eta \sum_{t \in T} \sum_{j=0}^{k-1} \nabla \overline{L}_i(\theta_{MT}^{(j)}) + \frac{\eta^2}{2} C(\{\theta_{MT}^{(j)}\}_{j=1}^{k-2}) + O(\eta^3) \tag{2}$$

with

$$C(\{\theta_{MT}^{(j)}\}_{j=1}^{h}) = \sum_{t \in T} \sum_{\ell=0}^{h} \nabla^2 \overline{L}_t(\theta_{MT}^{(\ell)}) \sum_{m=0}^{\ell} \left[ \alpha \sum_{t' \neq t, t' \in T} \nabla \overline{L}'_t(\theta_{MT}^{(m)}) + (\alpha - 1)\nabla \overline{L}_t(\theta_{MT}^{(m)}) \right]$$

We report the proof in appendix A.1. To better appreciate the relation between a task vector and a gradient computed over the corresponding task dataset, it is worth focusing on the single task case, in which one is exactly the additive inverse of the other, scaled by the learning rate $\eta$.

**Remark 3.1.** *From theorem 3.1, it follows that, for a single task $t$, and after a single epoch of finetuning,*

$$\tau_t = -\eta \nabla \overline{L}_t(\theta_{base})$$

*where $\eta$ is the learning rate.*

This also implies that, under the abovementioned assumptions, adding the task vector to the pre-trained model can be seen as an approximation of a finetuning of the latter.

**Corollary 3.1.1.** *Let $\theta_{TA}^{(k)}$ be the model obtained using vanilla task arithmetics i.e., $\theta_{TA}^{(k)} = \theta_{base} + \alpha \sum_{t=1}^{T} \tau_t^{(k)}$. Using the same notation of Theorem 3.1, it holds that*

$$\theta_{TA}^{(1)} = \theta_{MT}^{(1)} \tag{3}$$

$$\theta_{TA}^{(k)} = \theta_{MT}^{(k)} + \frac{\eta^2}{2} C(\{\theta_{MT}^{(j)}\}_{j=1}^{k-2}) + O(\eta^3) \quad \text{for } k \geq 2 \tag{4}$$

When $k > 1$, $\tau_{\text{MT}}$ still approximates the gradient of a model finetuned for the same number of epochs via gradient descent, but with an error that is $o(\eta^2)$. In fact, we show in fig. 2 that the multi-task model obtained by merging models finetuned for a single epoch works better than the one obtained by merging models finetuned to convergence. In other words, *the best resulting merged model is obtained when task arithmetic is exactly a step of gradient descent.* The effectiveness of task vectors in the $k > 1$ case, when the error nullifies the equality with respect to gradients, can instead be motivated by the observation that most of the trajectory followed by the model during the finetuning phase is dictated by the gradient of the first epoch: fig. 3a in fact shows that if we consider the epoch-wise normalized gradient norm for epoch $k$ obtained as

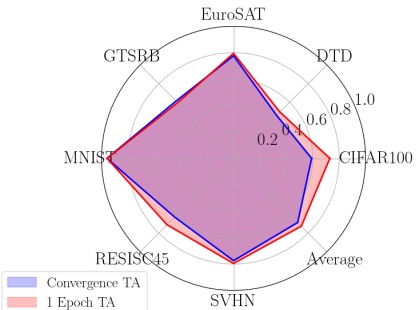

Figure 2: Accuracy of a model merged using task vectors over the task-specific models fine-tuned for 1 epoch and at convergence.

$$\nabla_k = \frac{\|\nabla_\theta^{(k)} L\|}{\sum_{k'=1}^{K} \|\nabla_\theta^{(k')} L\|},$$

then the first epoch accounts for the largest contribution, reaching, in some cases, almost 70% of the total of the gradient norms for all the epochs. When this is not true, e.g. in dataset RESISC45, we speculate that the direction is still mostly the one dictated by the previous epoch: looking at fig. 3b, we can see that the gradients of the first 5 epochs maintain high cosine similarity ($> 0.8$) with that of the first one. It is worth remarking that, by considering Gradient Descent (GD) instead of Stochastic Gradient Descent (SGD), the analysis does not apply exactly to the real case. However, considering SGD as an approximation to GD, we speculate that the intuition is still correct. Under this unifying lens, reducing task interference is analogous to reducing conflicting gradients in multi-task learning.

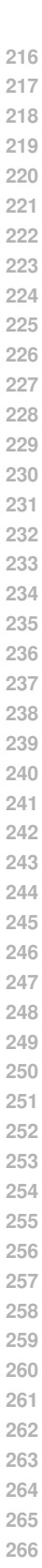
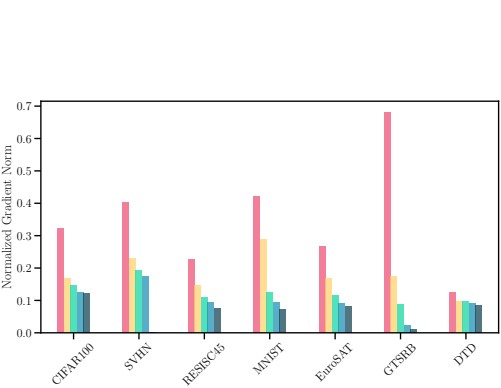

(a) Normalized gradient norms after 5 epochs of fine-tuning.

(b) Pairwise cosine similarities of the gradients of the first 10 epochs over dataset `RESISC45`.

## 4 ATM: ALTERNATING TUNING AND MERGING

Building upon the insights of section 3, we argue that task arithmetic is an approximation to a single GD step over the union of all the tasks. Following this parallel, we advocate taking further update steps sequentially and iteratively. The overall framework of ATM is depicted in fig. 1. Specifically, starting from a pretrained checkpoint as the base model $\theta_{\text{base}}^{(0)}$, we finetune it separately on each task to obtain the first-iteration task vectors $\tau_1^{(1)}, \ldots, \tau_{|T|}^{(1)}$. Task vectors are then aggregated and added to the base model to form the first-iteration unified model $\theta_{\text{base}}^{(1)}$. The procedure is iterated, and at each iteration $k$, the unified model $\theta_{\text{base}}^{(k)}$ becomes the new base model for the next iteration of ATM (eq. (5)). The $k$-th iteration task vector for task $t$, $\tau_t^{(k)}$, is obtained as $\theta_t^{(k)} - \theta_{\text{base}}^{(k)}$, where $\theta_t^{(k)}$ is a model obtained finetuning the $k$-th iteration base model $\theta_{\text{base}}^{(k)}$ on task $t$.

$$\theta_{\text{base}}^{(k+1)} = \theta_{\text{base}}^{(k)} + \frac{\alpha}{|T|} \sum_{t \in T} \tau_t^{(k)} \quad \forall k = 0, \ldots, K-1 \tag{5}$$

After $K$ iterations, we ultimately obtain $\theta_{\text{base}}^{(K)}$. The exact value of $K$ can be predefined or based on a stop condition. Using a pretrained model is the standard practice. However, Choshen et al. (2022) have suggested that pretrained checkpoints are not always the optimal starting point for further finetuning. Rather, they show that a model resulting from the merging of finetuned models can be a better starting point than any of its constituents. Inspired by this, throughout ATM iterations, we evolve the base model by iteratively merging task-specific finetuned models at each iteration until the compute budget is met. In practice, each iteration of ATM involves finetuning the current base model on all $|T|$ tasks of interest, thereby obtaining $|T|$ task vectors. These task vectors determine the task-specific directions the current base model should follow in order to attain enhanced performance on the corresponding tasks. The merging step of ATM at a given iteration simply consists of summing the mean current-iteration task vectors to the current base model, although any interference-resolution method in the task vector literature can be integrated. This step is intended to pull the base model closer to the multi-task basin on the loss landscape. The averaging step ensures the magnitude of the update remains insensitive to the number of tasks. Note that after each iteration, the task vectors of the previous iterations can be safely discarded. Therefore, at any instant of ATM, we only store one task vector for each of the $|T|$ tasks and the base model, incurring no additional memory requirements.

## 5 UPPER BOUNDING THE MULTI-TASK LOSS

In this section we explore the relationship between the the ATM loss, defined as the mean of average losses over all tasks, and the loss of a model trained jointly of all the datasets. Analogously to

section 3, we will conduct this analysis under the assumption that GD, rather than SGD, is used for optimizing the model parameters. This simplifying assumption removes the stochasticity introduced by random sampling, enabling a more straightforward analysis while still providing valuable insights into the underlying dynamics of the optimization process. With a slight abuse of notation, we denote with $t$ both the task and its corresponding dataset with cardinality $n_t$. The total number of samples for all tasks is given by $N = \sum_{t \in T} n_t$.

Inspired by the target model introduced by Daheim et al. (2023), we define as *target loss* for model merging the loss $L_{\text{target}}(\theta)$ of a model trained jointly on all the datasets. Namely, $L_{\text{target}}(\theta) = \frac{1}{N} \sum_{i=1}^{N} \ell(x_i, y_i, \theta)$. Given theorem 3.1, in the case in which we perform the merging after one step of finetuning on each dataset, the ATM update using gradient is

$$\theta_{\text{base}}^{(k+1)} = \theta_{\text{base}}^{(k)} + \frac{\alpha}{|T|} \sum_{t \in T} \tau_t^{(k)} = \theta_{\text{base}}^{(k)} - \alpha \eta \nabla \left( \frac{1}{|T|} \sum_{t \in T} \overline{L_t} \right)$$

Namely, we are performing a gradient descent step minimizing the function $L_{\text{ATM}} = \frac{1}{|T|} \sum_{t \in T} \overline{L_t}$. Having established that one step of ATM in gradient descent minimizes $L_{\text{ATM}}$, a crucial question arises: under what conditions does minimizing $L_{\text{ATM}}$ also lead to the minimization of $L_{\text{target}}$? In other words, when can we be certain that optimizing the ATM loss will also minimize the loss associated with training jointly on all datasets? To answer this question, we first note that $L_{\text{ATM}}$ is an unweighted average of the individual dataset losses, while the target loss is a weighted average:

$$L_{\text{target}}(\theta) = \frac{\sum_{t \in T} n_t \overline{L}_t(\theta)}{\sum_{t \in T} n_t},$$

We now analyze the parameter update from $\theta^{(k)}$ to $\theta^{(k+1)}$. For both ATM and target methods, we denote the change in loss, $L_{\text{method}}$, as $\Delta L_{\text{method}} = L_{\text{method}}(\theta^{(k)}) - L_{\text{method}}(\theta^{(k+1)})$. In the following theorem, we prove that if the drop in ATM loss exceeds a threshold $\delta$ the target loss will also decrease. The value of $\delta$ depends on the size of the largest dataset with a decreasing loss and the smallest dataset with an increasing loss. In particular, if the former dataset is larger than the latter, a reduction in ATM loss reduces the target loss. In practice, this is ensured when the loss is reduced on the largest dataset.

**Theorem 5.1.** *Let $D$ be the set of datasets where the loss decreases after a parameter update, and $I$ be the set of datasets where the loss increases or remains unchanged, defined as $D = \{t \mid \Delta \overline{L}_t > 0\}$ and $I = \{t \mid \Delta \overline{L}_t \leq 0\}$. If the reduction in the ATM loss satisfies $\Delta L_{ATM} > \delta$, where*

$$\delta = \frac{1}{|T|} \left( 1 - \frac{\min_{t \in I} n_t}{\max_{t \in D} n_t} \right) \sum_{t \in I} |\Delta \overline{L}_t|,$$

*then the target loss $L_{target}$ will also decrease, i.e., $\Delta L_{target} > 0$.*

We redirect the reader to appendix A.2 for the formal proof.

**Remark 5.1.** *If we choose the target loss to be the maximum of the average loss across all datasets $L_{target1} = \max_{t \in T} \overline{L}_t$, by leveraging the equivalence between the $L_1$-norm and the max norm, we obtain the bound $L_{target} \leq T \cdot L_{ATM}$.*

# 6 EXPERIMENTS

In this section, we illustrate the experimental setup and outcomes by comparing ATM against several recent baselines across a number of classification tasks in computer vision and NLP.

## 6.1 EXPERIMENTAL SETTING

**Datasets and Models**   To evaluate ATM, we conduct experiments across multiple neural architectures and datasets in both computer vision and natural language processing (NLP) domains. For computer vision tasks, we test ATM with a ViT-B-16 backbone (Dosovitskiy et al., 2021) and evaluate it on a diverse set of datasets: *CIFAR100* (Krizhevsky et al., 2009), *DTD* (Cimpoi et al., 2014),

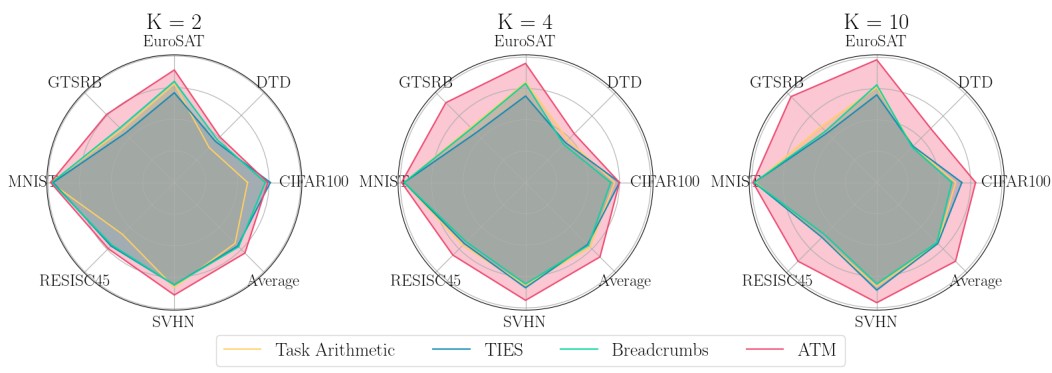

Figure 4: Comparisons as computational budget $K$ varies for ViT-B-16.

*EuroSAT* (Helber et al., 2019), *GTSRB* (Houben et al., 2013), *MNIST* (Lecun et al., 1998), *RE-SISC45* (Cheng et al., 2017), and *SVHN* (Netzer et al., 2011). For NLP tasks, we instead employ RoBERTa-base (Liu, 2019) and BERT-base-uncased (Devlin et al., 2019), evaluating them on the *GLUE benchmark* (Wang et al., 2019).

**Baselines and Metrics**   To gauge the performance of ATM, we consider several model merging baselines, including task arithmetic (TA) (Ilharco et al., 2022), TIES-merging (Yadav et al., 2023), and model breadcrumbs (Davari & Belilovsky, 2023) for both computer vision and NLP tasks, and DARE merging (Yu et al., 2023) for NLP tasks only. We adhere to author-recommended hyper-parameters, whenever needed or available, in order to ensure fair comparisons across experiments. Specifically, for TIES-merging, we retain the top $15\%$ of weights based on magnitude ranking. For model breadcrumbs, we set $\beta = 0.85$ and $\gamma = 0.993$. For DARE merging, we use a drop rate of 0.9. In all settings, we adopt mean aggregation of task vectors and use a scaling factor of 1 when applying them to the base model.

## 6.2 IMPACT OF EPOCH DISTRIBUTION ON PERFORMANCE

In this experiment, we first establish a fixed compute budget of 10 finetuning epochs for each task. Then, we seek the optimal distribution of epochs among different numbers of ATM iterations. To exemplify, if 10 epochs are distributed among 5 iterations, then in each iteration a task is finetuned for 2 epochs.

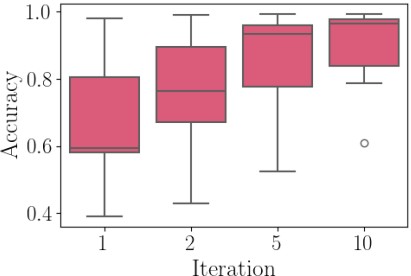

As shown in fig. 5, when a fixed compute budget is assumed, maximizing the number of iterations while minimizing the number of epochs per iteration achieves the best results for ATM. This suggests that applying more fine-grained updates to the base model is preferred to the application of abrupt updates. Distributing the budget of 10 epochs into 10 iterations achieves the highest average accuracy of 89%, surpassing the extreme of 1 iteration of

Figure 5: Multi-task accuracy for different budget distributions (ViT-B-16)

10 epochs by 21%. Note that this latter setting is analogous to task arithmetic, which performs one-shot merging. Following this finding, we use this 1 epoch, 10 iterations setting for most of the presented experiments.

## 6.3 EFFECT OF COMPUTE BUDGET

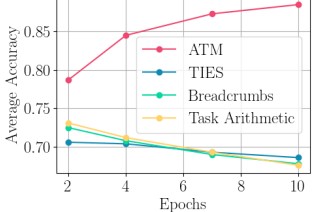

We extend the comparison of ATM against baseline methods under different compute budgets. Specifically, we vary the per-task number of finetuning epochs ($K$) within $\{2, 4, 10\}$, and compare the average test accuracy across tasks. As shown in 4, ATM's overall performance is remarkably superior, regardless of the budget.

7

Figure 6: Average multi-task accuracy as budget varies.

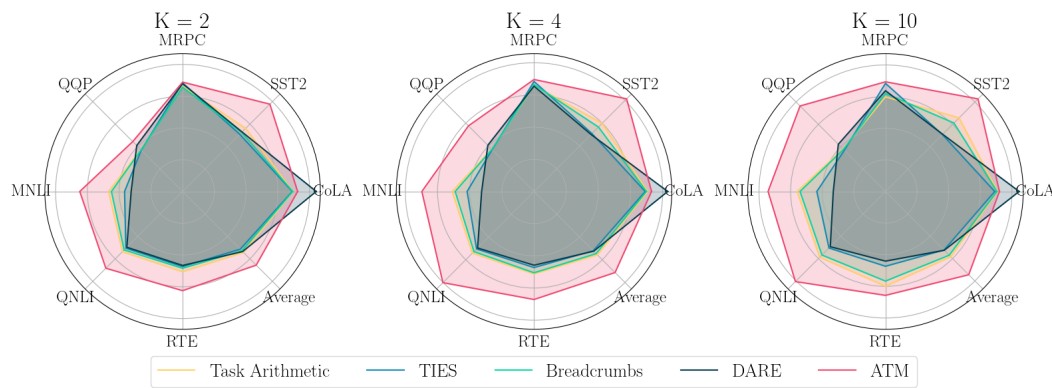

Figure 7: Comparisons as computational budget $K$ varies for RoBERTa-base.

Moreover, as the budget increases up to 10 epochs, the proportional advantage of ATM over baselines is also increased. ATM leads with an edge of 7%, 11%, and 21% over the best-performing baseline, for 2, 4, and 10 epochs of budget, respectively. For detailed results on both vision and NLP benchmarks, we refer the reader to B.2.

Interestingly, we observe that while the accuracy of ATM increases with more epochs of finetuning, the opposite is true for the baseline methods, see 6. In other words, *the more specialized the task-specific models, the lower the performance of the unified multi-task model*. ATM takes a different approach by gradually specializing the intermediate multi-task models, eschewing this issue.

## 6.4 COMPARISONS IN ORIGINAL SETTINGS

Assuming the availability of training data, we compare the following three ATM settings against various baselines under their original settings: (i) *ATM* finetuned on validation data (*valFT ATM*) for 10 iterations of 1 epoch, (ii) *ATM* finetuned on training data for 10 iterations of 1 epoch, and (iii) *ATM* finetuned on training data until convergence for 30 iterations of 1 epoch. As shown in table 1, all *ATM* variants outperform the baselines by a large margin. When the training data is available and the budget is limited to 10 epochs, ATM achieves an average accuracy of 89%, leading by 17% over the best-performing baseline. Assuming no compute budget restriction, ATM converges after 30 iterations, achieving a remarkable average test accuracy of 91%.

## 6.5 TRAINING-DATA FREE SETTING

One realistic constraint in practice is the absence of per-task training data, as finetuned models may be downloaded from an online repository. In this section, we assume only the availability of the validation data split, which is commonly exploited for hyperparameter tuning by baseline methods. In contrast to the baselines, in this setting, ATM uses the validation data for finetuning the various tasks, leaving hyperparameters untuned. We call this textitATM variant **valFT ATM**, and we compare its performance of against *ATM* and the baselines, adopting the author-suggested hyperparameters to ensure fair comparisons. As shown in Table 1, *valFT ATM* significantly outperforms the best-performing baseline by 10%. Note that the gap of 7% between valFT ATM and ATM is admittedly due to the amount of data for finetuning, but on all tasks (except for *EuroSAT*), valFT ATM performs comparably to ATM. We observe a similar phenomenon in the NLP domain as well.

## 6.6 TIME AND MEMORY COMPLEXITIES

This experiment compares ATM against baselines in terms of time and memory consumption, assuming the parameter count of the backbone to be $d$. Memory-wise, ATM introduces no additional requirements compared to task arithmetic, as the task vectors of each iteration can

| Method | Time | Memory |
|---|---|---|
| Task Arithmetic | $O(n * d)$ | $O(d)$ |
| TIES | $O(n * d * log(d))$ | $O(d)$ |
| Breadcrumbs | $O(n * d * log(d))$ | $O(d)$ |
| ATM | $O(k * n * d)$ | $O(d)$ |

Table 2: Comparison of methods in terms of time and memory complexity

| | CIFAR100 | DTD | EuroSAT | GTSRB | MNIST | RESISC45 | SVHN | Average |
|---|---|---|---|---|---|---|---|---|
| Task Arithmetic | 0.59 | 0.46 | 0.78 | 0.61 | 0.96 | 0.63 | 0.77 | 0.69 |
| TIES | 0.70 | 0.51 | 0.76 | 0.60 | 0.94 | 0.73 | 0.78 | 0.72 |
| Breadcrumbs | 0.60 | 0.47 | 0.77 | 0.60 | 0.95 | 0.63 | 0.75 | 0.68 |
| *ValFT ATM: 10 Orders* | 0.78 | 0.61 | 0.53 | 0.97 | 0.99 | 0.88 | 0.95 | 0.82 |
| *ATM: 10 Orders* | 0.79 | 0.61 | 0.98 | 0.97 | 0.99 | 0.89 | 0.96 | 0.89 |
| *ATM: 30 Orders* | **0.83** | **0.68** | **0.99** | **0.99** | **1.00** | **0.94** | **0.97** | **0.91** |

Table 1: Accuracy comparison under original baseline settings (ViT-B-16)

be deleted after the merge step. The time complexity of iteration-k ATM is equivalent to that of k times task arithmetic, as it iterates finetuning and merging for $k$ iterations. We argue that this time complexity is generally asymptotically negligible compared to those of TIES and breadcrumbs, of which the dominant time overhead stems from pruning each task vector post-hoc according to the magnitude ranking of all weights, incurring $O(d * log(d))$ time complexity. We compare the time and space complexities in table 2. Note that as long as $log(d)$ is asymptotically greater than $k$, which is generally the case since $k$ is usually a predefined constant integer (e.g. 10), ATM is faster than both TIES and breadcrumbs.

## 7 DISCUSSION

### 7.1 ORTHOGONALITY

Orthogonality between task vectors has been recommended as a desirable property for multi-task merging (Ilharco et al., 2022). Davari & Belilovsky (2023) adopts pairwise cosine similarity between task vectors as a proxy for task interference. Following this line, we again back the validity of this observation by identifying a positive correlation between ATM performance and task vector orthogonality. As shown in Figure fig. 9, as the number of ATM iterations increases, the magnitude of cosine similarity between task vectors tends to shrink, suggesting greater orthogonality as performance improves. Furthermore, as shown in 10, we find that ATM task vectors exhibit lower-magnitude average cosine similarity compared to the baseline methods.

### 7.2 TASK PROFICIENCY IS NOT MERGEABILITY

As shown in fig. 6, task-specific expertise does not imply multi-task performance. We observe that better-performing task-specific models result in worse multi-task models when adopting baseline methods, hinting that *downstream performance is not a predictor of post-merging performance*. We speculate that specialized models end up in highly dispersed locations in the parameter space, and merging them abruptly in a one-shot fashion generates a suboptimal multi-task model; this can be observed in fig. 8, where baseline methods end up all in the same (suboptimal) loss basin. On the contrary, a lower degree of specialization ensures the task-specific models remain closer to the pretrained checkpoint in the parameter space, leading to less aggressive updates when merging. The above insights translate to less aggressive updates (shorter-norm task vectors) being more amenable to merging. Capitalizing on this, ATM gradually aggregates task-specific models and updates the base model accordingly, merging less aggressively but over multiple iterations. At each iteration, the ATM task vectors represent the best nudges for the current base model without referring back to the initial pretrained model as all baseline methods do.

### 7.3 EDUCATED TRAJECTORY

Task arithmetic performs the aggregation step abruptly in a one-shot fashion over the initial pretrained checkpoint, likely overshooting the multi-task optimum. In ATM, however, the loss landscape is traversed iteration by iteration as the base model updates, leading to more informed nudges toward the multi-task optimum. Figure 8 depicts

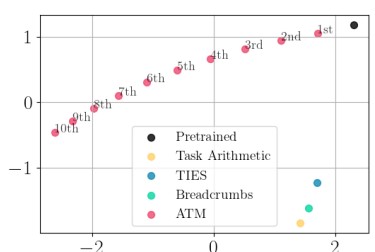

Figure 8: 2D PCA projection of various

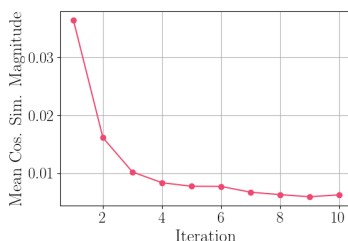

Figure 9: Average magnitudes of pairwise cosine similarity between ATM task vectors

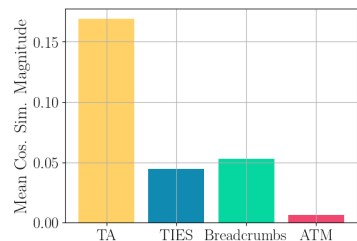

Figure 10: Average magnitudes of cosine similarity between task vectors

the 2D projection of various checkpoints via PCA. Notably, TIES and breadcrumbs, being post-hoc enhancements of task arithmetic, end up around the same basin, whereas ATM takes gradual steps toward a different and better basin, signaling the effectiveness of our novel iterative merging paradigm.

### 7.4 LIMITATIONS OF ITERATIVE MERGING

It is important to note that, while we have extensively shown the benefits of ATM in all the considered settings, its iterative nature also has its drawbacks. In particular, iterative merging does not yield a task representation that can be used for immediate adaptation to new tasks, as we instead sequentially obtain a series of $K$ vectors per task. storage. However, the approach is still advisable for obtaining a single model that performs best overall in all the tasks. In this regard, we made sure to maintain a fair comparison by using the same computing budget and data requirements, showing that a small validation set is sufficient to effectively employ the framework; such a set is always assumed to be presented in the literature and is usually used to optimize for the merging hyperparameters. Finally, while the approach is similar to performing gradient descent on the union of all the tasks, it still inherits one important property of task arithmetic: by obtaining the task vectors separately, the approach does not require centralizing the data on a computing node, making it amenable to federated settings where data privacy is a key requirement.

## 8 CONCLUSIONS

In conclusion, this paper identifies a key limitation of task arithmetic—overshooting due to abrupt model merging—and establishes its connection to gradient descent, forming the basis for our proposed model merging framework. We present Alternating Tuning and Merging (ATM), an iterative framework that addresses the shortcomings of one-shot merging techniques. By alternating between finetuning and merging, ATM effectively prevents overshooting and enhances multi-task performance. Extensive experiments on computer vision and NLP benchmarks demonstrate that ATM achieves state-of-the-art accuracy while maintaining computational efficiency comparable to existing baselines. Additionally, our theoretical analysis reveals that ATM optimizes the upper bound of the loss over the union of all the tasks and improves task vector orthogonality. The flexibility of ATM opens numerous future research directions, including the integration of interference-mitigation techniques and further refinement through advancements from the gradient descent literature.

### ETHICS STATEMENT

This research was conducted with a strong commitment to ethical standards in both data usage and experimental methodology. All datasets utilized in this study are publicly available. No personally identifiable information was accessed or used during the course of this research. Additionally, the experiments were designed to ensure fair comparisons across methods. We encourage future work that adheres to these same ethical principles and addresses broader societal impacts of machine learning technologies.

## REPRODUCIBILITY STATEMENT

We are committed to ensuring the reproducibility of our results and have taken steps to facilitate this for the broader research community. The code, datasets, and configurations used for the experiments in this paper are made available via a public repository. We are open to providing further instructions on the usage of our code. The hyperparameters, frameworks, and evaluation metrics have been thoroughly documented, and we provide clear descriptions of our experimental setup to allow for straightforward replication of our findings. We encourage the community to utilize these resources and provide feedback for further improvements.

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

# A PROOFS

## A.1 PROOF THEOREM 3.1 AND COROLLARY 3.1.1

In this section we will prove the Theorem 3.1 and Corollary 3.1.1. We repeat the statement of the theorem and of the corollary for easiness of reading.

**Theorem.** *3.1 Let $\left\{\theta_t^{(k)}\right\}_{t=1}^{|T|}$ be a set of models obtained by finetuning $\theta_{base}$ for $k$ epochs over the set of tasks $T$ via Gradient Descent (GD) with a learning rate of $\eta$, where finetuning over task $t \in T$ minimizes $\overline{L}_t(\theta) = \frac{1}{n_t} \sum_{i=1}^{n_t} \ell(x_i, y_i, \theta)$. Let moreover $\left\{\tau_t^{(k)}\right\}_{t=1}^{|T|}$ be a set of task vectors, with each $\tau_t^{(k)} = \theta_t^{(k)} - \theta_{base}$. Let $\tau_{MT}^{(k)}$ be the multi-task vector $\tau_{MT}^{(k)} = \sum_{t \in T} \tau_t^{(k)}$. Finally, let $\theta_{MT}^k$ be the model obtained minimizing the loss $\sum_{i=1}^{|T|} L_i$ for $k$ epochs. It holds that*

$$\tau_{MT}^1 = -\eta \nabla \sum_{t \in T} \overline{L}_t(\theta_{base}) \tag{6}$$

$$\tau_{MT}^k = -\eta \sum_{t \in T} \sum_{j=0}^{k-1} \nabla \overline{L}_i(\theta_{MT}^j) + \frac{\eta^2}{2} C(\{\theta_{MT}^j\}_{j=1}^k) + O(\eta^3) \tag{7}$$

*with*

$$C(\{\theta_{MT}^j\}_{j=1}^k) = \sum_{t \in T} \sum_{\ell=0}^k \nabla^2 \overline{L}_t(\theta_{MT}^\ell) \sum_{m=0}^\ell \left[ \alpha \sum_{t' \neq t, t' \in T} \nabla \overline{L}'_t(\theta_{MT}^m) + (\alpha - 1) \nabla \overline{L}_t(\theta_{MT}^m) \right]$$

**Corollary.** *3.1.1 Let $\theta_{TA}^k$ be the model obtained using vanilla task arithmetics i.e., $\theta_{TA}^k = \theta_{base} + \alpha \sum_{t=1}^{|T|} \tau_t^k$. It holds that*

$$\theta_{TA}^1 = \theta_{MT}^1 \tag{8}$$

$$\theta_{TA}^k = \theta_{MT}^k + \eta^2 o(1) \quad \text{for } k > 1 \tag{9}$$

*where $\eta$ is the learning rate used for the finetuning of the model on the single datasets.*

We recall that $\theta_i^k$ is the model obtained finetuing on task $i$ for $k$ epochs and that both the finetuing on different tasks and the training on the average loss start from a pretrained model $\theta^0$.

To prove the statement of the theorem and of the corollary we need a intermediate result. We introduce the following notation:

$$r_i(\theta) = \alpha \sum_{j \neq i} \nabla \overline{L}_j(\theta) + (\alpha - 1) \nabla \overline{L}_i(\theta) = \alpha \sum_{j=1}^{|T|} \nabla \overline{L}_j(\theta) - \nabla \overline{L}_i(\theta^{(0)}) \tag{10}$$

$$p_i^k(\theta^0, \theta_{MT}^1, \ldots, \theta_{MT}^k) = \sum_{j=0}^k r_i(\theta_{MT}^k) \tag{11}$$

$$s_i^k(\theta^0, \ldots, \theta_{MT}^k) = \sum_{j=0}^k \nabla^2 \overline{L}_i(\theta_{MT}^j)[p_i^j(\theta^0, \ldots, \theta_{MT}^{j-1})]. \tag{12}$$

**Lemma A.1.** *Using the notation introduced in theorem 3.1, it holds that*

$$\theta_i^{(1)} = \theta_{MT}^{(1)} + \eta p_i^0(\theta^0) \tag{13}$$

*and for $m \geq 2$*

$$\theta_i^{(m+1)} = \theta_{MT}^{(m+1)} + \eta p_i^m(\theta^0, \ldots, \theta_{MT}^m) - \frac{\eta^2}{2} s_i^{m-1}(\theta^0, \ldots, \theta_{MT}^{m-1}) + O(\eta^3) \tag{14}$$

*Proof.* We first show that the statement is true for $m = 1$, and then prove the results for $m \geq 2$ by induction. In this case, the base case is given for $m = 2$. In the induction step, instead, we prove that if the statement holds for any given case $m$ then it must also hold for the next case $m + 1$.

$m = 1$**. First epoch** For each task $i = 1, \ldots, |T|$

$$\theta_i^{(1)} = \theta^{(0)} - \eta \nabla \overline{L}_i(\theta^{(0)}) \text{ while } \theta_{MT}^{(1)} = \theta^{(0)} - \alpha \eta \sum_{i \in T} \nabla \overline{L}_i(\theta^{(0)}).$$

Consequently, it holds that

$$\theta_i^1 = \theta_{MT}^{(1)} + \eta \left[ \alpha \sum_{j \neq i} \nabla \overline{L}_j(\theta^{(0)}) + (\alpha - 1) \nabla \overline{L}_i(\theta^{(0)}) \right]$$

$$= \theta_{MT}^{(1)} + \eta r_i(\theta^{(0)}) = \theta_{MT}^{(1)} + \eta p_i^0(\theta^0).$$

$m = 2$. **Second epoch**

$$\theta_i^{(2)} = \theta_i^1 - \eta\nabla\overline{L}_i(\theta_i^1)$$

$$= \theta_{\text{MT}}^{(1)} + \eta r_i(\theta^{(0)}) - \eta\nabla\overline{L}_i\left(\theta_{\text{MT}}^{(1)} + \eta r_i(\theta^{(0)})\right)$$

$$\overset{Taylor}{\approx} \theta_{\text{MT}}^{(1)} + \eta r_i(\theta^{(0)}) - \eta\nabla\overline{L}_i(\theta_{\text{MT}}^{(1)}) - \frac{\eta^2}{2}\nabla^2\overline{L}_i(\theta_{\text{MT}}^1)r_i(\theta^{(0)}) + O(\eta^3)$$

$$= \theta_{\text{MT}}^{(1)} - \eta\nabla\overline{L}_i(\theta_{\text{MT}}^{(1)}) + \eta r_i(\theta^{(0)}) - \frac{\eta^2}{2}\nabla^2\overline{L}_i(\theta_{\text{MT}}^1)r_i(\theta^{(0)}) + O(\eta^3)$$

$$= \theta_{\text{MT}}^{(1)} - \eta\nabla\overline{L}_i(\theta_{\text{MT}}^{(1)}) + \eta\alpha\sum_{t\in T}\nabla\overline{L}_i(\theta_{\text{MT}}^{(1)}) - \eta\alpha\sum_{t\in T}\nabla\overline{L}_i(\theta_{\text{MT}}^{(1)}) + \eta r_i(\theta^{(0)})$$

$$- \frac{\eta^2}{2}\nabla^2\overline{L}_i(\theta_{\text{MT}}^1)r_i(\theta^{(0)}) + O(\eta^3)$$

$$= \theta_{\text{MT}}^{(1)} + \eta r_i(\theta_{\text{MT}}^1) + \eta r_i(\theta^{(0)}) - \frac{\eta^2}{2}\nabla^2\overline{L}_i(\theta_{\text{MT}}^1)r_i(\theta^{(0)}) + O(\eta^3)$$

$$= \theta_{MT}^1 + \eta p_i^1(\theta^0,\ldots,\theta_{\text{MT}}^1) - \frac{\eta^2}{2}s_i^0(\theta^0) + O(\eta^3)$$

**Inductive step** Let us assume that

$$\theta_i^m = \theta_{MT}^m + \eta p_i^{m-1}(\theta^0,\ldots,\theta_{\text{MT}}^{m-1}) - \frac{\eta^2}{2}s_i^{m-2}(\theta^0,\ldots,\theta_{\text{MT}}^{m-2}) + O(\eta^3)$$

We can derive that

$$\theta_i^{m+1} = \theta_i^m - \eta\nabla\overline{L}_i(\theta_i^m)$$

$$= \theta_{MT}^m + \eta p_i^{m-1}(\theta^0,\ldots,\theta_{\text{MT}}^{m-1}) - \frac{\eta^2}{2}s_i^{m-2}(\theta^0,\ldots,\theta_{\text{MT}}^{m-2}) - \eta\nabla\overline{L}_i(\theta_i^m) + O(\eta^3)$$

$$= \theta_{MT}^m + \eta p_i^{m-1}(\theta^0,\ldots,\theta_{\text{MT}}^{m-1}) - \frac{\eta^2}{2}s_i^{m-2}(\theta^0,\ldots,\theta_{\text{MT}}^{m-2})$$

$$- \eta\nabla\overline{L}_i\left(\theta_{MT}^m + \eta p_i^{m-1}(\theta^0,\ldots,\theta_{\text{MT}}^{m-1}) - \frac{\eta^2}{2}s_i^{m-2}(\theta^0,\ldots,\theta_{\text{MT}}^{m-2})\right) + O(\eta^3)$$

$$= \theta_{MT}^m + \eta p_i^{m-1}(\theta^0,\ldots,\theta_{\text{MT}}^{m-1}) - \frac{\eta^2}{2}s_i^{m-2}(\theta^0,\ldots,\theta_{\text{MT}}^{m-2})$$

$$- \eta\nabla\overline{L}_i(\theta_{MT}^m) - \frac{\eta}{2}\nabla^2\overline{L}_i(\theta_{MT}^m)\left(\eta p_i^{m-1}(\theta^0,\ldots,\theta_{\text{MT}}^{m-1}) - \frac{\eta^2}{2}s_i^{m-2}(\theta^0,\ldots,\theta_{\text{MT}}^{m-2})\right) + O(\eta^3)$$

$$= \theta_{MT}^m + \eta p_i^{m-1}(\theta^0,\ldots,\theta_{\text{MT}}^{m-1}) - \frac{\eta^2}{2}s_i^{m-2}(\theta^0,\ldots,\theta_{\text{MT}}^{m-2})$$

$$- \eta\nabla\overline{L}_i(\theta_{MT}^m) - \frac{\eta^2}{2}\nabla^2\overline{L}_i(\theta_{MT}^m)p_i^{m-1}(\theta^0,\ldots,\theta_{\text{MT}}^{m-1}) + O(\eta^3)$$

$$= \theta_{MT}^{(m+1)} + \eta p_i^m(\theta^0,\ldots,\theta_{\text{MT}}^m) - \frac{\eta^2}{2}s_i^{m-1}(\theta^0,\ldots,\theta_{\text{MT}}^{m-1}) + O(\eta^3)$$

$\square$

*Proof Theorem and Corollary.* For the first epoch

$$\theta_{\text{TA}}^1 = \theta^0 + \alpha\sum_{i\in T}\tau_i^1 = \theta^0 - \eta\alpha\sum_{i\in T}\nabla\overline{L}_i(\theta^0)$$

while, choosing $\alpha\eta$ as learning rate for the loss $\sum_{i\in T}\overline{L}_i$ :

$$\theta_{\text{MT}}^{(1)} = \theta^{(0)} - \alpha\eta\sum_{i\in T}\nabla\overline{L}_i(\theta^{(0)}).$$

So $\theta_{\text{MT}}^{(1)} = \theta_{\text{TA}}^1$.

For $k \geq 2$, notice that

$$\theta_{MT}^k = \theta^0 - \alpha\eta \sum_{j=0}^{k-1} \nabla \sum_{t\in T} \overline{L}_i(\theta_{MT}^j). \tag{15}$$

Now, using Lemma A.1 , using that

$$-\alpha\eta \sum_{j=0}^{k-1} \nabla \sum_{t\in T} \overline{L}_t(\theta_{MT}^j) + \eta p_i^{k-1}(\eta^0, \ldots, \eta_{\text{MT}}^{k-1})$$

$$= -\alpha\eta \sum_{j=0}^{k-1} \nabla \sum_{t\in T} \overline{L}_t(\theta_{MT}^j) + \sum_{j=0}^{k-1} r_i(\theta_{\text{MT}}^k)$$

$$= -\alpha\eta \sum_{j=0}^{k-1} \nabla \sum_{t\in T} \overline{L}_i(\theta_{MT}^j) + \sum_{j=0}^{k-1} \alpha \sum_{j\in T} \nabla\overline{L}_j(\theta_{MT}^j) - \nabla\overline{L}_i(\theta_{MT}^j)$$

$$= -\eta \sum_{j=0}^{k-1} \nabla\overline{L}_i(\theta_{MT}^j)$$

Namely

$$\theta_i^{m+1} = \theta_{MT}^{(m+1)} + \eta p_i^m(\theta^0, \ldots, \theta_{\text{MT}}^m) - \frac{\eta^2}{2} s_i^{m-1}(\theta^0, \ldots, \theta_{\text{MT}}^{m-1}) + O(\eta^3)$$

$$= \theta^0 - \alpha\eta \sum_{j=0}^{m} \nabla \sum_{t\in T} \overline{L}_i(\theta_{MT}^j) + \eta p_i^m(\theta^0, \ldots, \theta_{\text{MT}}^m) - \frac{\eta^2}{2} s_i^{m-1}(\theta^0, \ldots, \theta_{\text{MT}}^{m-1}) + O(\eta^3)$$

$$= \theta^0 - \eta \sum_{j=0}^{m} \nabla\overline{L}_i(\theta_{MT}^j) - \frac{\eta^2}{2} s_i^{m-1}(\theta^0, \ldots, \theta_{\text{MT}}^{m-1}) + O(\eta^3)$$

we can rewrite the tasks vectors as

$$\tau_i^k = \theta_i^k - \theta^0 \tag{16}$$

$$= -\eta \sum_{j=0}^{k-1} \nabla\overline{L}_i(\theta_{MT}^j) - \frac{\eta^2}{2} s_i^{k-2}(\theta^0, \ldots, \theta_{\text{MT}}^{k-2}) + O(\eta^3) \tag{17}$$

Consequently the model obtain with TA is

$$\theta_{\text{TA}}^k = \theta^0 + \alpha \sum_{i\in T} \tau_i^k$$

$$= \theta^0 - \eta\alpha \sum_{j=0}^{k-1} \sum_{i\in T} \nabla\overline{L}_i(\theta_{MT}^j) - \alpha \sum_{i\in T} \frac{\eta^2}{2} s_i^{k-2}(\theta^0, \ldots, \theta_{\text{MT}}^{k-2}) + O(\eta^3)$$

$$= \theta_{\text{MT}}^k - \alpha \sum_{i\in T} \frac{\eta^2}{2} s_i^{k-2}(\theta^0, \ldots, \theta_{\text{MT}}^{k-2}) + O(\eta^3)$$

$$\square$$

## A.2   PROOF THEOREM 5.1

In this section, we provide a rigorous proof of Theorem 5.1, establishing the conditions under which ATM implicitly optimizes the loss of the model trained on the union of datasets.

*Proof.* Suppose that when transitioning from parameter $\theta^i$ to parameter $\theta^{i+1}$, the change in the average loss for each dataset $D_k$ is given by $\Delta \overline{L_k} = \overline{L_k}(\theta^i) - \overline{L_k}(\theta^{i+1})$. We denote by $P$ the set of tasks for which the delta of the loss is positive and by $P$ the set for which it is negative, namely $P = \{k \in \mathrm{T}\ \text{s.t.}\ \Delta \overline{L_k} > 0\}$ and $N = \{k \in \mathrm{T}\ \text{s.t.}\ \Delta \overline{L_k} \le 0\}$ The set $\{1, \ldots, n\} = P \cup N$. In the following formulas we will use the symbol $|$ different purposes. For sets, like $D_i$, $|D_j|$ denotes the cardinality of the set, while for scalars, such as $\Delta \overline{L_k}$, $|\Delta \overline{L_k}|$ denote their absolute value. Since the task in $N$ have negative $\Delta \overline{L_k}$, it holds that

$$\sum_{j \in N} |D_j| \Delta \overline{L_j} = - \sum_{j \in N} |D_j| |\Delta \overline{L_j}|.$$

For hypothesis $\Delta L_{\mathrm{ATM}} > \delta$, this implies $\sum_{j=1}^{n} \Delta_j > n\delta$. We have that $\sum_{j \in P} \Delta \overline{L_j} + \sum_{j \in N} \Delta \overline{L_j} = \sum_{j \in P} \Delta \overline{L_j} - \sum_{j \in N} |\Delta \overline{L_j}| > n\delta$, namely $\sum_{j \in P} \Delta \overline{L_j} > n\delta + \sum_{j \in N} |\Delta \overline{L_j}|$

We want to prove that $\Delta L_{\mathrm{target}} > 0$. Let us now consider $\Delta L_{\mathrm{target}} > 0$ iff $\sum_{j=1}^{n} |D_j| \Delta_j > 0$.

$$\begin{aligned}
\sum_{j=1}^{n} |D_j| \Delta_j &= \sum_{j \in P} |D_j| \Delta \overline{L_j} + \sum_{j \in N} |D_j| \Delta \overline{L_j} \\
&= \sum_{j \in P} |D_j| \Delta \overline{L_j} - \sum_{j \in N} |D_j| |\Delta \overline{L_j}| \\
&> \min_{j \in P} |D_j| \sum_{j \in P} \Delta \overline{L_j} - \max_{j \in N} |D_j| \sum_{j \in N} |\Delta \overline{L_j}| \\
&> \min_{j \in P} |D_j| \left[ n\delta + \sum_{j \in N} |\Delta \overline{L_j}| \right] - \max_{j \in N} |D_j| \sum_{j \in N} |\Delta \overline{L_j}| \\
&= n \min_{j \in P} |D_j| \delta + (\min_{j \in P} |D_j| - \max_{j \in N} |D_j| \sum_{j \in N}) |\Delta \overline{L_j}|
\end{aligned}$$

The last line of the previous equation is positive by hypothesis since we assumed $\delta > \frac{1}{n}(1 - \frac{\min_{j \in N} |D_j|}{\max_{j \in P} |D_j|}) \sum_{j \in N} |\Delta \overline{L_j}|$. $\qquad\qquad\square$

# B  ADDITIONAL RESULTS

## B.1  FULL RESULTS OVER VARYING COMPUTATIONAL BUDGET

In the main paper, we pictorially depicted the multi-task accuracies of the baselines and variants of ATM, in the form of radar plots. For deeper analysis, here we report the full results of ATM compared to the baselines, as the computational budget varies for 2, 4, 7, and 10 epochs.

## B.2  COSINE SIMILARITY OF EPOCH-WISE GRADIENTS

We report in fig. 11 the cosine similarity of gradients for the first 10 epochs over DTD and EuroSAT, as these do not have a marked difference in gradient norms between the first epoch and the remaining ones in fig. 3a. We see that the alignment of subsequent gradients observed in RESISC45 still holds in DTD, even if with less marked similarities. On the other hand, this does not seem to hold for EuroSAT.

| | | CIFAR100 | DTD | EuroSAT | GTSRB | MNIST | RESISC45 | SVHN | average |
|---|---|---|---|---|---|---|---|---|---|
| $K = 2$ | Task Arithmetic | 0.58 | 0.39 | 0.78 | 0.59 | 0.98 | 0.58 | 0.83 | 0.68 |
| | TIES | **0.76** | 0.46 | 0.71 | 0.55 | 0.96 | 0.71 | 0.81 | 0.71 |
| | Breadcrumbs | 0.72 | 0.48 | 0.80 | 0.61 | 0.97 | 0.70 | 0.81 | 0.72 |
| | ValFT ATM | 0.70 | **0.53** | 0.84 | **0.84** | **0.98** | **0.78** | **0.90** | **0.80** |
| | ATM | 0.74 | 0.51 | **0.89** | 0.76 | **0.98** | 0.74 | 0.89 | 0.79 |
| $K = 4$ | Task Arithmetic | 0.71 | 0.48 | 0.80 | 0.62 | 0.97 | 0.71 | 0.82 | 0.73 |
| | TIES | 0.75 | 0.45 | 0.69 | 0.56 | 0.97 | 0.69 | 0.84 | 0.70 |
| | Breadcrumbs | 0.68 | 0.43 | 0.79 | 0.61 | 0.97 | 0.67 | 0.81 | 0.71 |
| | ValFT ATM | 0.66 | **0.55** | 0.73 | 0.85 | 0.98 | **0.83** | **0.94** | 0.79 |
| | ATM | **0.75** | **0.55** | **0.95** | **0.90** | **0.99** | 0.82 | **0.94** | **0.84** |
| $K = 7$ | Task Arithmetic | 0.67 | 0.44 | 0.79 | 0.62 | 0.97 | 0.67 | 0.83 | 0.71 |
| | TIES | 0.72 | 0.43 | 0.66 | 0.56 | 0.97 | 0.66 | 0.85 | 0.69 |
| | Breadcrumbs | 0.65 | 0.41 | 0.76 | 0.62 | 0.97 | 0.62 | 0.81 | 0.69 |
| | ValFT ATM | 0.75 | **0.62** | 0.66 | **0.96** | **0.99** | 0.85 | **0.96** | 0.83 |
| | ATM | **0.77** | 0.58 | **0.98** | **0.96** | **0.99** | **0.87** | 0.95 | **0.87** |
| $K = 10$ | Task Arithmetic | 0.63 | 0.41 | 0.76 | 0.63 | 0.97 | 0.62 | 0.83 | 0.69 |
| | TIES | 0.68 | 0.41 | 0.70 | 0.56 | 0.98 | 0.62 | 0.86 | 0.69 |
| | Breadcrumbs | 0.60 | 0.40 | 0.78 | 0.58 | 0.98 | 0.59 | 0.81 | 0.68 |
| | ValFT ATM | 0.78 | **0.61** | 0.53 | **0.97** | **0.99** | 0.88 | 0.95 | 0.82 |
| | ATM | **0.79** | **0.61** | **0.98** | **0.97** | **0.99** | **0.89** | **0.96** | **0.89** |

Table 3: ATM vs. Baselines as budget varies (*ViT-B-16*)

| | | CoLA | SST2 | MRPC | QQP | MNLI | QNLI | RTE | Average |
|---|---|---|---|---|---|---|---|---|---|
| $K = 2$ | Task Arithmetic | 0.70 | 0.56 | 0.66 | 0.37 | 0.47 | 0.54 | 0.51 | 0.54 |
| | TIES | 0.69 | 0.51 | 0.68 | 0.37 | 0.37 | 0.51 | 0.47 | 0.51 |
| | Breadcrumbs | 0.69 | 0.53 | 0.66 | 0.37 | 0.45 | 0.52 | 0.48 | 0.53 |
| | DARE | **0.84** | 0.53 | 0.68 | 0.41 | 0.32 | 0.50 | 0.47 | 0.54 |
| | valFT ATM | 0.71 | **0.78** | 0.68 | **0.46** | 0.64 | **0.68** | **0.67** | **0.66** |
| | ATM | 0.72 | **0.78** | **0.69** | 0.44 | **0.65** | **0.68** | 0.62 | **0.66** |
| $K = 4$ | Task Arithmetic | 0.71 | 0.60 | 0.66 | 0.37 | 0.51 | 0.54 | 0.51 | 0.56 |
| | TIES | 0.69 | 0.51 | 0.68 | 0.37 | 0.42 | 0.51 | 0.47 | 0.52 |
| | Breadcrumbs | 0.70 | 0.57 | 0.66 | 0.37 | 0.49 | 0.53 | 0.51 | 0.55 |
| | DARE | **0.83** | 0.50 | 0.65 | 0.41 | 0.33 | 0.50 | 0.46 | 0.53 |
| | valFT ATM | 0.70 | **0.83** | **0.70** | **0.68** | 0.67 | 0.73 | 0.62 | 0.70 |
| | ATM | 0.73 | 0.81 | **0.70** | 0.58 | **0.70** | **0.80** | **0.67** | **0.71** |
| $K = 7$ | Task Arithmetic | 0.71 | 0.65 | 0.65 | 0.38 | 0.52 | 0.56 | 0.55 | 0.57 |
| | TIES | 0.69 | 0.51 | 0.68 | 0.37 | 0.42 | 0.51 | 0.47 | 0.52 |
| | Breadcrumbs | 0.71 | 0.61 | 0.65 | 0.37 | 0.50 | 0.54 | 0.52 | 0.56 |
| | DARE | **0.84** | 0.52 | 0.65 | 0.43 | 0.33 | 0.49 | 0.44 | 0.52 |
| | valFT ATM | 0.69 | **0.85** | 0.69 | 0.75 | 0.68 | 0.72 | 0.60 | 0.71 |
| | ATM | 0.72 | 0.79 | **0.70** | 0.73 | **0.73** | **0.82** | **0.67** | **0.74** |
| $K = 10$ | Task Arithmetic | 0.71 | 0.66 | 0.60 | 0.38 | 0.56 | 0.59 | 0.60 | 0.59 |
| | TIES | 0.69 | 0.51 | 0.68 | 0.37 | 0.43 | 0.51 | 0.47 | 0.52 |
| | Breadcrumbs | 0.71 | 0.61 | 0.62 | 0.38 | 0.54 | 0.57 | 0.57 | 0.57 |
| | DARE | **0.85** | 0.51 | 0.64 | 0.42 | 0.33 | 0.49 | 0.44 | 0.52 |
| | valFT ATM | 0.68 | **0.86** | 0.66 | 0.76 | 0.70 | 0.73 | 0.63 | 0.72 |
| | ATM | 0.72 | 0.83 | **0.69** | 0.76 | **0.74** | **0.81** | **0.66** | **0.74** |

Table 4: ATM vs Baselines as budget varies (*RoBERTa-base*)

|  |  | CoLA | SST2 | MRPC | QQP | MNLI | QNLI | RTE | Average |
|---|---|---|---|---|---|---|---|---|---|
| $K = 2$ | Task Arithmetic | 0.61 | 0.70 | 0.39 | 0.65 | 0.41 | 0.57 | 0.54 | 0.55 |
|  | TIES | 0.32 | 0.50 | 0.32 | 0.63 | 0.37 | 0.50 | 0.54 | 0.45 |
|  | Breadcrumbs | 0.60 | 0.68 | 0.39 | 0.65 | 0.40 | 0.57 | 0.54 | 0.55 |
|  | DARE | **0.82** | 0.54 | **0.69** | 0.40 | 0.33 | 0.50 | 0.46 | 0.53 |
|  | ValFT ATM | 0.66 | **0.71** | 0.47 | 0.65 | 0.44 | **0.57** | 0.56 | **0.58** |
|  | ATM | 0.68 | 0.68 | 0.39 | 0.64 | **0.46** | 0.56 | **0.61** | **0.58** |
| $K = 4$ | Task Arithmetic | 0.59 | 0.66 | 0.39 | 0.67 | 0.41 | 0.55 | **0.57** | 0.55 |
|  | TIES | 0.32 | 0.50 | 0.32 | 0.63 | 0.38 | 0.51 | 0.53 | 0.46 |
|  | Breadcrumbs | 0.58 | 0.65 | 0.40 | 0.67 | 0.41 | 0.55 | 0.56 | 0.54 |
|  | DARE | **0.82** | 0.53 | **0.66** | 0.43 | 0.33 | 0.50 | 0.47 | 0.54 |
|  | ValFT ATM | 0.67 | **0.75** | 0.47 | **0.70** | 0.54 | **0.64** | 0.62 | 0.62 |
|  | ATM | 0.68 | 0.76 | 0.44 | 0.74 | **0.59** | 0.66 | **0.67** | **0.65** |
| $K = 7$ | Task Arithmetic | 0.58 | 0.65 | 0.38 | 0.66 | 0.42 | 0.55 | 0.58 | 0.55 |
|  | TIES | 0.33 | 0.50 | 0.33 | 0.63 | 0.38 | 0.51 | 0.53 | 0.46 |
|  | Breadcrumbs | 0.58 | 0.65 | 0.38 | 0.66 | 0.41 | 0.55 | 0.59 | 0.55 |
|  | DARE | **0.83** | 0.52 | **0.66** | 0.43 | 0.33 | 0.51 | 0.46 | 0.53 |
|  | ValFT ATM | 0.66 | 0.75 | 0.47 | 0.70 | 0.59 | **0.67** | 0.64 | 0.64 |
|  | ATM | 0.69 | **0.81** | 0.51 | **0.78** | **0.63** | 0.70 | 0.66 | **0.68** |
| $K = 10$ | Task Arithmetic | 0.59 | 0.66 | 0.39 | 0.64 | 0.42 | 0.55 | **0.59** | 0.55 |
|  | TIES | 0.35 | 0.51 | 0.34 | 0.64 | 0.38 | 0.51 | 0.53 | 0.47 |
|  | Breadcrumbs | 0.59 | 0.65 | 0.39 | 0.64 | 0.42 | 0.55 | 0.60 | 0.55 |
|  | DARE | **0.82** | 0.51 | **0.69** | 0.42 | 0.33 | 0.51 | 0.47 | 0.54 |
|  | ValFT ATM | 0.66 | **0.73** | 0.50 | **0.71** | 0.59 | **0.69** | 0.63 | 0.64 |
|  | ATM | 0.68 | 0.81 | 0.53 | 0.78 | **0.65** | 0.72 | **0.66** | **0.69** |

Table 5: ATM vs. Baselines as budget varies (*BERT-base-uncased*)

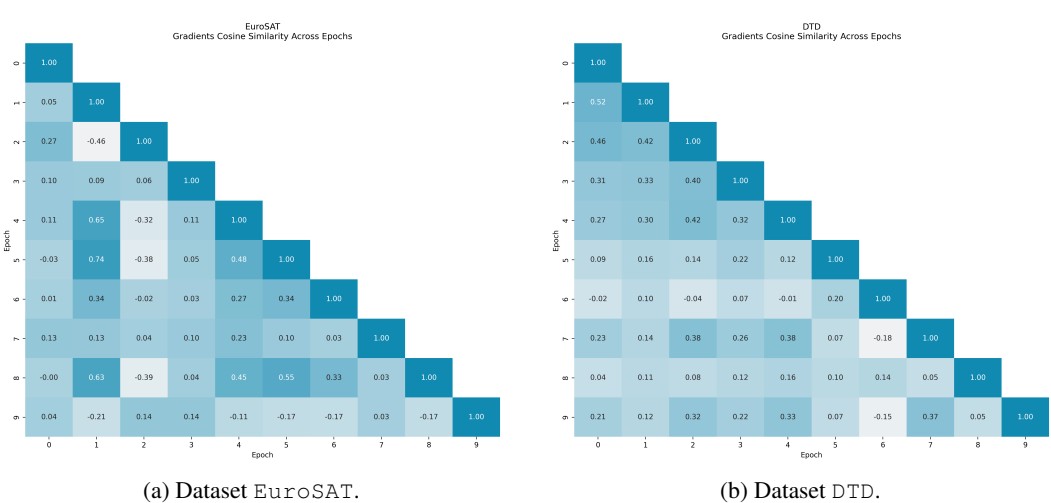

(a) Dataset `EuroSAT`.  (b) Dataset `DTD`.

Figure 11: Pairwise cosine similarities of the gradients of the first 10 epochs over datasets that do not exhibit most of the gradient norm localized in the first epoch.

