# OpenReview forum: "ATM: Improving Model Merging by Alternating Tuning and Merging"
_ICLR.cc/2025/Conference — ICLR 2025 Conference Withdrawn Submission_

### Official Review · Reviewer_SmWb · 2024-11-01

**Soundness:** 3
**Presentation:** 3
**Contribution:** 3
**Rating:** 5
**Confidence:** 4

**Summary:**

In this paper, the authors first discover through theory and experiment that merging models of fine-tuning until convergence is not the optimal choice. Consequently, they introduce a technique that involves independently training each model for one epoch and then using model merging to create a new model to serve as the starting point for the next training round. This process is repeated to obtain the final merged model. The authors conducted experiments on models such as ViT-B-16 and achieved remarkable results.

**Strengths:**

1. **Theoretical Support**: The authors use proven theories in some discussions to support their viewpoints.
2. **Comprehensive Ablation Study**: There is discussion and experimental verification of various aspects of the proposed method.

**Weaknesses:**

1. **Scope of the Method and Application Scenario**: The proposed method requires original training data (at least validation data) and training, whereas the baselines (including Task Arithmetic, TIES, DARE, etc.) used for comparison are in fact data-free and training-free merging methods, allowing them to directly merge existing models. The requirements for data and training are obligatory for the authors’ method, thus making the application scenarios quite different. The authors should compare their method with others that require similar amounts of data and training (at least with the most standard MTL results). Alternatively, authors could define the approach as a new training paradigm that utilizes merging method, rather than simply as a model merge method.
2. **Inappropriate Cost Comparison**: For the same reason, the comparison of computational and storage costs in Section 6.6 is also unreasonable. The merging stage of the authors’ method remains Task Arithmetic, but it combines with training to change the timing of merging. Thus, it is unreasonable to only consider the computational and storage costs of the merging stage; training costs are unavoidable.

**Questions:**

1. When observing Figure 6 and Table 1 together, is the baseline result reported in Table 1 corresponding to the result at epoch 10? Although the authors' method achieves the best results at 10 or even 30 epochs, is 10 epochs too many for these single-task fine-tuning tasks?
2. In Figure 6, the authors illustrate, through the declining curve of baselines like Task Arithmetic, the expert capability increased with the decreased performance of the final merged model. However, could the performance decline of the merged model be due to over-training of individual expert models? It is suggested that the authors report the performance changes of a single expert model across epochs to further support their claim.
3. Regarding valFT ATM, reaching such astonishing results with only the validation set is noteworthy. Could the authors provide more information on this setting?
4. In my understanding, Theorem 3.1 and Corollary 3.1.1 conclude that models obtained by GD then merged and the model obtained by MTL loss (defined as \bar{L}_t(\theta) in Theorem 3.1) are somewhat equivalent only for 1 epoch. When exceeding 1 epoch, additional errors are introduced (not sure if I understood correctly?). I strongly suggest the authors report the results of the standard MTL (simultaneously fine-tuning multiple tasks) for further theoretical validation.

Typos:
Formatting errors in various figures and tables, including Figures 6 and 8 and Table 2.
The caption of Figure 3 is missing.

---

### Official Review · Reviewer_14SC · 2024-11-03

**Soundness:** 1
**Presentation:** 1
**Contribution:** 1
**Rating:** 1
**Confidence:** 5

**Summary:**

The paper proposes *Alternating Tuning and Merging* (ATM), a novel approach for merging single-task models into a multi-task model. ATM alternates between fine-tuning each task separately and merging through weight interpolation techniques, such as task arithmetic or TIES. Experimental results show ATM’s advantages over standard model merging baselines for small ViT models on computer vision tasks and BERT-like models on an NLP benchmark.

**Strengths:**

1. The paper addresses model merging, an emerging area of high relevance for reducing fine-tuning costs in multi-task settings.

2. The initial theoretical insights linking task vectors with gradients of the loss for the corresponding tasks after the first gradient descent step are novel.

**Weaknesses:**

1. **Misalignment with Model Merging Goals**. The paper appears to overlook the key aims of model merging. The motivation for merging models fine-tuned on different tasks is not only saving memory – as the introduction of the paper claims – but instead bypassing expensive joint fine-tuning by enabling the combination of models fine-tuned *independently* on separate tasks. Indeed, model merging methods often rely on available checkpoints from online hubs. Contrary to this goal, ATM’s merging process mixes models already at the fine-tuning stage. This loses all the modularity properties and undermines the very benefits of model merging. Consequently, the baseline most relevant for this study is standard multi-task learning, i.e., joint fine-tuning, which is missing (and expected to obtain superior performance over ATM). Table 2 comparing computational costs is therefore misleading since ATM requires joint fine-tuning. For example, adding one task with Task Arithmetic or TIES is straightforward. Adding one task with ATM requires full retraining of the model on all tasks.

2. **Baseline Comparison and Parameter Choices**. The baselines used in the experiments rely on suboptimal hyperparameters (see, e.g., line 350 "in all settings, we adopt mean aggregation of task vectors and use a scaling factor of 1 when applying them to the base model"). This decision severely weakens the validity of all results of the paper and prevents a fair comparison with past approaches. Moreover, the paper cites but omits evaluation against state-of-the-art merging methods such as AdaMerging, which, despite its significant computational overhead, is far less resource-intensive than ATM.

3. **Experimental Scope**. The experiments are limited to small and/or outdated models. For instance, in the case of computer vision, only the ViT B16 model is used. Standard benchmark architectures such as ViT large, T5, or LLaMA models are not included. Arguably, this is due to the excessive computational overhead of ATM compared to present methods, which hinders the scalability of ATM, further reinforcing the concerns above.

4. **Presentation of Results**. The paper opts for qualitative radar charts over numerical tables, complicating performance comparisons. Single-task accuracies after standard fine-tuning are never reported for the baselines, normalized accuracies are also missing. Together with the use of suboptimal hyperparameters and a choice of datasets that are not the standard in the field, this makes it difficult to compare the results against previous work.

5. **Theoretical Analysis and Testing**. The theoretical analysis lacks soundness. Firstly, the fact that task vectors are precisely equal to single-task gradient updates just for the first step of gradient descent does not imply that the best merging performance is obtained when merging after the first step. In fact, the accuracy of the model also improves in the next steps, so even if errors accumulate, this does not exclude the improvement of performance after merging. Secondly, the paper does not test this hypothesis directly with gradient descent; instead, it uses stochastic gradient descent, for which the analysis does not hold. On the one hand, the natural approximation is to approximate one step of GD with one step of SGD (with large batch size) in the early phases of training and not with a full epoch consisting of hundreds if not thousands of steps. On the other hand, testing it with gradient descent is straightforward using gradient accumulation.  If done, results would likely show that merging performance post first step is not optimal, as the model would still closely reflect zero-shot capabilities at this stage.

6. **Presentation Quality**. The paper suffers from several presentation issues. Figures are of low quality, extend beyond page margins, and are occasionally truncated. Some captions are missing. Table 2 is presented before Table 1, ignoring the logical flow. Additionally, the paper’s notation is inconsistent, e.g., symbols such as $\theta_0, \theta_{base}, \theta_{init}$ appear to refer to the same quantity.

**Questions:**

None.

---

### Official Review · Reviewer_4aJH · 2024-11-03

**Soundness:** 1
**Presentation:** 1
**Contribution:** 1
**Rating:** 1
**Confidence:** 5

**Summary:**

The paper proposes a method that replaces the finetune-and-then-merge paradigm with alternating fine-tuning and merging. The paper also connects task vectors with multi-task gradients and evaluates on standard vision and NLP benchmarks.

**Strengths:**

The proposed method has a GPU memory advantage compared to gradient-balancing multi-task learning approaches, since the fine-tuning for each task can be performed independently.

**Weaknesses:**

The paper has many flaws, from poor placement in the literature, misconfiguring baselines and ignoring the most relevant ones, overstating the results and representing known results as novel. Specifically:


1. The method is wrongly presented as model merging, while it is more closely related to joint fine-tuning. Imo, it cannot be seen as model merging, since the task combination is known a priori. The method is inflexible and defeats the purpose of model merging
2. Moreover, there is not a single reference for multi-task learning, nor is it considered as a baseline. The method mostly resembles gradient-based MTL solutions [1-3, inter alia], where instead of gradients being computed at every step, task vectors are computed and merged after some steps. This can actually be implied by lines 88-91. The paper would benefit from looking into this perspective by first motivating the need of gradient-balancing solutions (vs loss-balancing [4])in the fine-tuning paradigm (which might actually not be the case) and then formulating the method as a way to combat the memory issues associated with gradient-balancing
3. Task vectors vs gradients: this is an obvious connections. For one step of (S)GD, the definitions are the same.
4. Lack of proper citations: no mention of multi-task literature as well as other places. For instance: “under this unifying lens, reducing task interference is analogous to reducing conflicting gradients in multi-task learning” misses [5,6] for interference and [3] for conflicting gradients.
5. Section 6 is poorly structured, starting with ablation studies culminating into the main results deep into the section
6. Lack of scientific rigor: “In all setting, we adopt mean aggregation of task vectors and use a scaling factor of 1 when applying them to the base model”. This shows that the baselines are misconfigured, making the comparison unfair.
7. Overstating the results: Given the lack of proper baseline selection and correct setup of used baselines, the stated improvements of “up to 19% in computer vision and 20% in NLP over the best baselines” are far-fetched.
8. “greatest contribution is given by the first epoch”: this is presented as a novel insight but it is not. This occurs naturally during training since the model adapts to the new distribution and, the same applies to Fine-tuning
9. Increased orthogonality: the comparison is unfair, since the authors choose to compare the task vectors of the last iteration with the “general” ones form standard fine-tuning. They will be more orthogonal since the joint features are already embedded in the multi-task vector via the proposed joint fine-tuning and the last fine-tuning only focuses on task-specific features.

Given the above, I disagree with all four of the stated contributions of this work.

[1] Navon A, Shamsian A, Achituve I, Maron H, Kawaguchi K, Chechik G, Fetaya E. Multi-task learning as a bargaining game. ICML 2022.

[2] Chen Z, Badrinarayanan V, Lee CY, Rabinovich A. Gradnorm: Gradient normalization for adaptive loss balancing in deep multitask networks. InInternational conference on machine learning 2018 Jul 3 (pp. 794-803). PMLR.

[3] Yu T, Kumar S, Gupta A, Levine S, Hausman K, Finn C. Gradient surgery for multi-task learning. Advances in Neural Information Processing Systems. 2020;33:5824-36.

[4] Kendall, Alex, Yarin Gal, and Roberto Cipolla. "Multi-task learning using uncertainty to weigh losses for scene geometry and semantics." Proceedings of the IEEE conference on computer vision and pattern recognition. 2018.

[5] Yadav, P., Tam, D., Choshen, L., Raffel, C.A. and Bansal, M., 2024. Ties-merging: Resolving interference when merging models. *Advances in Neural Information Processing Systems*, *36*.

[6] Wang, K., Dimitriadis, N., Ortiz-Jimenez, G., Fleuret, F. and Frossard, P., 2024. Localizing Task Information for Improved Model Merging and Compression. *ICML 2024*.

**Questions:**

1. How is section 5 connected to the method specifically? It seems to be restating multi-task learning findings

---

### Official Review · Reviewer_rFsT · 2024-11-04

**Soundness:** 3
**Presentation:** 2
**Contribution:** 2
**Rating:** 5
**Confidence:** 4

**Summary:**

This paper proposes a novel model “merging” algorithm, where the goal is to merge separately tuned, task-specific models into a joint multi-task model. The paper begins by making several observations regarding the paradigm of task-specific fine-tuning followed by model merging. First, they demonstrate the relationship between fine-tuning then model merging and standard multi-task learning (i.e. following the average task gradient at each step). Additionally, they show that merging the model after a single epoch results in stronger generalization than merging after the task-specific models converge on their respective tasks, and present some empirical analysis to help explain this finding.

Motivated by these findings, the authors propose Alternating Tuning and Merging (ATM), wherein training alternates between making task-specific updates to task-specific models and making a “global” update by merging the task-specific models together. The authors demonstrate that ATM significantly outperforms other model merging baselines, which all consist of a single merge step. Additionally, the authors provide analysis of performance regarding the frequency at which models are merged vs. tuned under an epoch budget, showing that merging more frequently improves performance.

**Strengths:**

The proposed method (ATM) is fairly simple, but is shown to have significantly higher performance over “one-shot” baselines and methods, where models are fine-tuned once and then merged. This demonstrates the value of merging models intermittently, rather than training models fully in a task-specific manner and then merging them.

The paper also has some interesting analysis regarding the gradients of individual tasks over the course of training, which can help explain the efficacy of ATM and help us understand why merging earlier as opposed to later can be beneficial.

**Weaknesses:**

My understanding is that model merging is primarily a technique for taking advantage of models which are fine-tuned separately from one another, e.g. combining separate open-source models trained by different individuals. In other words, model merging is a technique to construct multi-task models when multi-task data is not available, i.e. when a single joint model cannot be trained. The proposed method, however, assumes that we have access to the entire multi-task data at one time, in order to alternate between tuning and merging.

In this case, it is not clear to me why we would not simply do multi-task learning, and indeed this paper seems to suggest that we should be doing precisely that: performance goes up as we merge more often during training, and as we merge more often during training we more closely approximate multi-task learning. However, pure MTL (i.e. no task-specific updates) is not considered here as a baseline. Thus, it is not clear to me how this new method (ATM) fits into the current paradigm of model merging or transfer learning, outside of an analysis on the impact of the rate at which models are merged versus tuned.

It is also worth mentioning that the proposed method is similar to heterogeneous federated averaging, where learning algorithms often alternate between making local “task-specific” updates to task-specific models and making global “multi-task” updates using changes to the task-specific models. In this literature, it has been established that “global SGD”, which makes no local updates, has better convergence guarantees but that “local SGD” which makes some task-specific updates before merging can have some generalization benefits [1,2]. I feel strongly that this literature needs to be acknowledged and discussed in this paper, given their similarities.

A smaller comment: in line 420 it is claimed that the hyperparameters are untuned in the data-free setting, and then later mentioned that the author-suggested hyperparameters are used. I am not sure that both can be true however, as it is likely the case that the previous authors used the validation data for tuning the hyperparameters, and therefore that the hyperparameters actually have seen the data being used for training in this case.

The floating figures need to be fixed, there are 3 pages where the figure hangs off the bottom edge of the page and messes up the formatting of the next page as well.

[1] Woodworth et al., 2022; Minibatch vs Local SGD for Heterogeneous Distributed Learning
[2] Bao et al., 2024; Provable Benefits of Local Steps in Heterogeneous Federated Learning for Neural Networks: A Feature Learning Perspective

**Questions:**

See weaknesses.

---

### Note · Authors · 2024-11-24

**Comment:**

We would like to thank the reviewers for their detailed feedback and valuable insights. After considering the reviews, we recognize the need to better position our method within the existing literature, especially concerning multi-task learning and model merging. We plan to address these issues and improve the theoretical analysis in a future submission. Thank you for your time and constructive critique.

**Withdrawal Confirmation:**

I have read and agree with the venue's withdrawal policy on behalf of myself and my co-authors.